# Blood Cell-Derived Inflammatory Indices in Diabetic Macular Edema: Clinical Significance and Prognostic Relevance

**DOI:** 10.3390/biomedicines13122979

**Published:** 2025-12-04

**Authors:** Chiyu Lin, Weiqing Ye, Suyao Wu, Zijing Huang

**Affiliations:** 1Joint Shantou International Eye Center of Shantou University and The Chinese University of Hong Kong, Shantou 515041, China; 24cylin@stu.edu.cn (C.L.); 23wqye@stu.edu.cn (W.Y.); 24sywu@stu.edu.cn (S.W.); 2Fifth Clinical Institute, Shantou University Medical College, Shantou 515041, China

**Keywords:** diabetic macular edema, inflammation, neutrophil-to-lymphocyte ratio, platelet-to-lymphocyte ratio, systemic immune-inflammation index, biomarkers, prognosis

## Abstract

Diabetic macular edema (DME) is a leading cause of vision loss in patients with diabetes. While VEGF-driven vascular permeability is central to its pathogenesis, inflammation plays a complementary and pivotal role in disease progression, morphological heterogeneity, and treatment response. Readily available blood cell-derived inflammatory indices, such as the neutrophil-to-lymphocyte ratio (NLR), platelet-to-lymphocyte ratio (PLR), systemic immune-inflammation index (SII), monocyte-to-high-density lipoprotein cholesterol ratio (MHR), monocyte-to-lymphocyte ratio (MLR), platelet-to-neutrophil ratio (PNR), and pan-immune-inflammation value (PIV), as well as platelet measures (MPV, PDW), have been investigated as low-cost markers of systemic inflammation in DME. Specifically, comparative studies have reported that an NLR ≥ 2.26 can effectively distinguish DME from non-DME with 85% sensitivity and 74% specificity. Elevated NLR is more associated with serous retinal detachment. Moreover, a baseline NLR ≤ 2.32 has been linked to a better anatomical response to treatment. This narrative review summarizes the evidence regarding these biomarkers’ diagnostic and prognostic utility and highlights their associations with OCT morphotypes and anti-VEGF responsiveness. We propose that multi-marker panels integrated with OCT features may enhance risk stratification and help personalize therapy, but emphasize that prospective, multi-center validation and harmonized thresholds are required before routine clinical application.

## 1. Introduction

Diabetic retinopathy (DR) and diabetic macular edema (DME) remain major causes of vision impairment worldwide [1]. Recent global estimates indicate a rising burden of DR and DME driven by aging populations and increasing diabetes prevalence [2,3,4]. DME is the most frequent cause of vision loss among patients with diabetes, and carries a significant risk of irreversible visual decline if untreated [5]. Current first-line therapy, intravitreal anti-vascular endothelial growth factor (VEGF) injections, improves vision for many patients but leaves a substantial proportion (approximately 40%) with persistent fluid or recurrence, and imposes a high follow-up burden [1,6,7]. These clinical gaps motivate the search for accessible biomarkers that predict disease activity, morphological phenotype, and treatment response.

The pathogenesis of DR and DME is multifactorial. Among them, inflammation is increasingly recognized as a driver of DME pathophysiology. Inflammatory mediators, including cytokines (IL-1β, IL-6, TNF-α), chemokines (MCP-1), complement fragments, and adhesion molecules (ICAM-1, VCAM-1), contribute to blood–retinal barrier (BRB) disruption, leukostasis, and altered fluid dynamics [8,9,10,11]. Importantly, peripheral blood counts and derived ratios provide inexpensive surrogates of systemic immune status and inflammatory activity, which are emerging as accessible indices for predicting disease severity and treatment outcomes, especially for anti-VEGF therapy [12,13,14,15].

## 2. Brief Overview of Diabetic Macular Edema

DME results from the abnormal accumulation of intraretinal or subretinal fluid due to BRB dysfunction and altered retinal fluid homeostasis [16]. Chronic hyperglycemia drives multiple pathogenic pathways, including polyol flux, advanced glycation end-product formation, protein kinase C system, oxidative stress, and the local renin–angiotensin system, leading to endothelial dysfunction, retinal pigment epithelium (RPE) impairment, and upregulation of VEGF and proinflammatory cytokines [17,18,19,20,21,22,23,24,25]. Activated retinal microglia and recruited monocyte-derived macrophages amplify local inflammation and vascular permeability [26,27].

Based on OCT features, DME presents with three common morphologies: (1) diffuse retinal thickening (DRT)—retinal thickening (>250 µm) with reduced intraretinal reflectivity and enlarged hyporeflective areas; (2) cystoid macular edema (CME)—intraretinal cystoid spaces appearing as round or oval hyporeflective cavities separated by hyperreflective septa; (3) serous retinal detachment (SRD)—subretinal accumulation of optically empty fluid beneath a hyperreflective, dome-shaped detached neurosensory retina [28]. These morphotypes reflect distinct anatomic patterns and, increasingly, differing inflammatory milieus.

Systemic risk factors for DME include poor glycemic control, hypertension, dyslipidemia, longer diabetes duration, diabetic nephropathy, and severe retinopathy. Specifically, each 1% increase in glycated hemoglobin (HbA1c) confers a 10–25% higher risk, while a 10-mmHg reduction in systolic blood pressure reduces risk by about 15% [29,30,31]. While systemic control reduces incidence and progression, it only explains a small element of individual risk variation, highlighting the need for additional biomarkers to refine individual prognosis.

## 3. Current Therapeutic Landscape

Anti-VEGF agents are the mainstay of DME treatment and effectively reduce macular thickness and improve visual acuity in many patients [32]. However, a considerable proportion of patients with DME exhibit poor response to early anti-VEGF therapy. Accumulating evidence indicates that elevated baseline levels of blood cell-derived inflammatory indices, such as the neutrophil-to-lymphocyte ratio (NLR), platelet-to-lymphocyte ratio (PLR), and systemic immune-inflammation index (SII), may predict a poorer response to anti-VEGF therapy and a potentially better response to steroids [33,34]. Corticosteroid implants target inflammatory pathways and are useful especially in steroid-responsive but anti-VEGF-refractory cases; however, their use is constrained by cataract progression and intraocular pressure elevation [35,36]. Pars plana vitrectomy remains an option for tractional or vitreomacular interface-related disease [32]. For refractory DME or persistent DME after PPV, micropulse laser can partially improve macular edema and visual function without the adverse effects associated with conventional laser therapy [37,38]. Systemic anti-inflammatory therapies are not routinely recommended due to limited evidence and systemic adverse effects [39].

## 4. Inflammation in DME: Mechanistic Links to Biomarkers

Chronic hyperglycemia induces metabolic stress in retinal glia, endothelial cells, and RPE, triggering the release of cytokines (e.g., IL-6, IL-1β, TNF-α) [9,40,41], chemokines (MCP-1) [42], matrix metalloproteinases (MMP-9), and complement components [43,44]. These mediators reduce tight-junction protein expression, promote leukocyte adhesion and transmigration, and increase vascular permeability, thereby driving blood–retinal barrier (BRB) disruption and fluid accumulation. Cytokine signaling pathways such as NF-κB, STAT3, and MAPK sustain this low-grade inflammatory state [8,45,46,47,48] and can be amplified by pro-angiogenic factors (notably VEGF), which further potentiate vascular leakage [44,48,49,50].

Systemic inflammation both mirrors and may exacerbate intraocular inflammation; activated circulating leukocytes, platelet reactivity, and dyslipidemia create a proinflammatory, pro-thrombotic milieu that plausibly contributes to BRB disruption and fluid accumulation. Consequently, peripheral indices that capture neutrophil, monocyte, lymphocyte, and platelet kinetics provide inexpensive, indirect readouts of the underlying pathobiology. The immune axes and mechanistic pathways associated with these inflammatory indices are summarized in Appendix A.

## 5. Blood Cell-Derived Inflammatory Indices in DME

Blood cell-derived inflammatory indices are cost-effective surrogates of systemic inflammation and have been evaluated as potential biomarkers of DME activity and treatment response. Below, we summarize the most studied indices. For each marker, we outline its definition, key cross-sectional and longitudinal findings, associations with OCT morphotypes where available, and practical strengths and limitations. Table 1 summarizes the current research reports and main conclusions on blood cell-derived inflammatory indices for DME detection and discrimination of OCT-based subtypes, and Table 2 summarizes the current research reports and key findings on these indices for predicting anti-VEGF treatment response in DME patients. Figure 1 demonstrates the peripheral blood cell populations and associated inflammatory mediators captured by different blood-derived inflammatory indices, as well as their potential relevance to inflammation-related macular diseases.

### 5.1. White Blood Cell-Related Inflammatory Indices

#### 5.1.1. Neutrophil-to-Lymphocyte Ratio (NLR)

NLR has been most extensively studied in DME. Compared with diabetic patients without DR (1.89 ± 0.70), NLR increases in those with DR but no DME (2.22 ± 1.51, *p* < 0.001) and is highest in patients with DME (3.40 ± 2.42, *p* < 0.001 vs. both groups). Several studies report thresholds (≈2.0–2.4) that discriminate DME from non-DME diabetes with moderate accuracy. For instance, one study suggests an NLR threshold of 2.0 for discriminating DME, with 75% sensitivity and 59% specificity (AUC 0.72) [33], while another report indicates an optimal cutoff of ≥2.26, showing 85% sensitivity and 74% specificity [51]. Higher NLR was significantly associated with the coexistence of DME (*p* = 0.002) [52]. In addition, NLR differs among optical coherence tomography (OCT) subtypes of DME, with higher NLR in patients with CME [13] and SRD [12,64] compared with DRT.

NLR also serves as a potential prognostic biomarker. Higher baseline NLR has been associated with greater central retinal thickness (CRT) after anti-VEGF therapy; however, its association with functional outcomes (BCVA) is inconsistent. Hu et al. identified pre-treatment NLR ≥ 2.27 as an independent predictor of poor visual outcomes following anti-VEGF therapy [60]. Lower NLRs have been observed in anti-VEGF responders compared with non-responders [60]. Consistently, Ergin et al. demonstrated that an NLR of <2.32 was associated with ≥10% CRT reduction one month after anti-VEGF treatment [34]. Elevated baseline NLR (≥2.31) has also been associated with higher rates of significant visual improvement after dexamethasone implantation [61]. Some studies have used the reciprocal of NLR, namely the lymphocyte-to-neutrophil ratio (LNR), for evaluation. For example, Karimi et al. found that higher LNR was associated with better visual improvement after anti-VEGF treatment [62].

The strengths of NLR include its accessibility, low cost, and robust association with both DME onset and treatment outcomes. However, its predictive value for visual functional outcomes is inconsistent. Furthermore, NLR is a nonspecific marker and may be influenced by a range of physiological or pathological conditions beyond DME, such as infection, systemic inflammation, and medications. Combining NLR with other indices or OCT features improves predictive performance.

#### 5.1.2. Platelet-to-Lymphocyte Ratio (PLR)

PLR has emerged as another widely studied inflammation-related index following NLR. Current studies consistently report elevated PLR levels in patients with DME. One study demonstrated a PLR of 117.9 ± 59.1 in DME patients compared to 106.3 ± 28.7 in diabetic patients without DME [12]. Similarly, Mohamed et al. reported similar findings and, through ROC analysis, suggested that PLR provides incremental diagnostic value for differentiating DME from non-DME diabetes [53]. In addition, Gündoğdu et al. observed higher PLR values in DME cases with SRD compared to uncomplicated DME, although the difference did not reach statistical significance [12]. Cheng et al. further summarized that in DME patients with SRD, PLR levels were markedly elevated and correlated with imaging severity, thereby supporting the role of PLR as a component of multimarker inflammatory profiling [64].

Regarding functional and anatomical outcomes, PLR has also demonstrated predictive value. For example, Hu et al. identified baseline PLR as an independent negative predictor of BCVA improvement after anti-VEGF therapy [60]. Katić [15] and Ergin [34] similarly reported significantly higher baseline PLR in DME non-responders compared with responders, indirectly supporting the notion that elevated PLR reflects a more inflammatory DME phenotype. PLR was also predictive of CRT reduction following treatment, although its diagnostic accuracy was only moderate (AUC 0.63) [15,60]. Some studies have used the reciprocal of PLR, namely the lymphocyte-to-platelet ratio (LPR), for evaluation. For example, Karimi et al. found that higher LPR was associated with better visual improvement after anti-VEGF treatment; however, their analysis failed to show an association between LPR and CRT reduction, suggesting that LPR may be more closely linked to functional rather than anatomical outcomes [62].

The substantial inter-study variability in cut-off values and the overlap with comorbid conditions, such as cardiovascular disease and malignancies, limit the specificity of PLR as an inflammatory biomarker. Moreover, PLR is susceptible to the modulatory effects of antiplatelet therapy, which may modestly attenuate circulating platelet counts or perturb platelet production dynamics without necessarily exerting a concomitant impact on retinal inflammatory activity. Consequently, PLR values may decline in treated individuals even when systemic or intraocular inflammatory status remains unchanged, thereby introducing a potential confounding variable when comparing PLR between DME cohorts and other study populations.

#### 5.1.3. Systemic Immune-Inflammation Index (SII)

The SII, calculated as platelet × neutrophil/lymphocyte, has emerged as a composite biomarker reflecting thrombotic, innate, and adaptive immune responses. Current evidence consistently indicates elevated SII levels in patients with DME, underscoring its close association with disease activity. One study reported a mean SII of 599.7 ± 279.2 in DME eyes compared with 464.9 ± 172.2 in diabetic controls without DME [12]. Among DME cases, patients with SRD exhibited significantly higher SII values than those with uncomplicated DME [12,64]. This evidence suggests a stepwise rise in SII alongside retinal fluid accumulation, supporting its role as a surrogate marker of systemic inflammation linked to BRB disruption.

The association between SII and both functional and anatomical outcomes has been well established. For instance, a baseline SII below 543.53 showed a greater reduction in central macular thickness after 30 days of anti-VEGF therapy [34]. In addition, baseline SII was markedly higher in anti-VEGF non-responders (675.3 ± 334.0) compared with responders (445.3 ± 166.3) [15]. Chen et al. incorporated SII into a nomogram and identified a cut-off value of 531.23 (AUC 0.613) for predicting visual prognosis. When combined with OCT biomarkers such as SRF and hyperreflective foci, the discrimination improved substantially (AUC 0.866) [63]. Importantly, these associations remained significant after adjustment for age, sex, diabetes duration, and CRP, highlighting the robustness of SII as a prognostic marker. Recently, a prospective study further confirmed that baseline SII was significantly higher in anti-VEGF non-responders compared with responders (886 ± 407 vs. 547 ± 183), with ROC analysis demonstrating an AUC of 0.788 for predicting anatomical response, superior to NLR and PLR [34].

The advantage of SII lies in its integration of platelet-driven thrombo-inflammation and neutrophil-mediated oxidative stress. However, reported cut-off values vary considerably (approximately 400–600), reflecting heterogeneity across studies. This heterogeneity may be attributed to differences in study populations, ethnic backgrounds, comorbidities, DME morphologic subtypes, DR severity, anti-VEGF regimens, and response criteria across studies. Variability in sampling conditions and laboratory methods may also affect SII measurement. These factors can influence SII distribution and limit the reproducibility of a universal cut-off value across diverse clinical settings. In addition, similar to PLR, antiplatelet therapy may influence SII through mechanisms beyond platelet count. These drugs attenuate platelet activation and suppress P-selectin-mediated adhesion, thereby inhibiting the formation of platelet–neutrophil and platelet–monocyte aggregates that amplify systemic inflammation [65]. Such immunomodulatory effects could reduce SII values even when the underlying inflammatory burden of DME remains unchanged. However, direct evidence in this specific clinical context remains limited, and this interpretation should be considered mechanistically plausible but speculative.

#### 5.1.4. Monocyte-to-High-Density Lipoprotein Cholesterol Ratio (MHR)

The MHR has recently gained attention as a composite index linking innate immune activation with antioxidant reserve. One study reported that the MHR in the eyes of patients with DME (15.7 ± 5.7) was significantly higher than in diabetic patients without DME (13.2 ± 7.5) and in healthy controls (12.5 ± 3.9). ROC analysis identified a threshold of 13.9, which predicted DME with 81% specificity, superior to the specificity of NLR ≥ 2 (59%) [33]. However, MHR was also found to show no significant difference between patients with and without DME [54]. In addition, MHR demonstrated limited prognostic relevance for anatomical or functional outcomes. No significant associations were observed between baseline MHR and CRT or BCVA at presentation. In addition, MHR failed to predict changes in CRT and BCVA following anti-VEGF therapy, suggesting that MHR is not a reliable short-term marker of treatment response [33,54].

#### 5.1.5. Monocyte-to-Lymphocyte Ratio (MLR)

The MLR levels are significantly higher in individuals with DME compared with diabetic individuals without DME and healthy controls, suggesting its potential as a biomarker for assessing disease severity and prognosis. In a study by Mohamed et al., a progressive and statistically significant rise in MLR was found, from healthy controls (0.153) to diabetic eyes without DME (0.157) and to eyes with DME (0.169, IQR) [53]. In addition, different DME subtypes exhibit distinct MLR levels. Liao et al. demonstrated that the CME group had significantly higher MLR than the DRT group [13], while Sun et al. observed that DME patients with SRD had higher MLR than other DME subtypes [55]; however, they did not find a significant difference between the DRT and CME groups. Further research is required to clarify the threshold values and diagnostic accuracy of MLR across different DME subtypes.

Evidence regarding the predictive role of MLR in treatment response remains heterogeneous. MLR has been identified as an independent predictor of early anatomical response to anti-VEGF therapy [60]. Ergin et al. report that baseline MLR was significantly higher in patients who failed to achieve an early anatomical response after intravitreal anti-VEGF therapy for diabetic macular edema [58]. Mani et al. similarly suggested that MLR could predict anatomical changes, although it was not correlated with functional visual improvement after treatment [52]. In contrast, Vural et al. reported that patients with high baseline MLR (≥0.30) achieved a higher proportion of ≥1-line visual improvement following intravitreal dexamethasone implantation [61]. One study introduced the reciprocal of MLR, namely the lymphocyte-to-monocyte ratio (LMR). However, they reported no significant association between the LMR and either visual improvement or CRT reduction following anti-VEGF therapy [62].

#### 5.1.6. Platelet-to-Neutrophil Ratio (PNR)

The PNR has recently been investigated as a systemic biomarker for DME. A cohort study demonstrated a stepwise decrease in PNR: 100.66 in healthy controls, 92.39 in diabetics without DR, 95.63 in DR eyes without DME, and 50.73 in eyes with DME. A threshold of PNR ≤ 68.51 was identified as an independent risk factor for DME, with a diagnostic sensitivity of 80.2% and specificity of 75.6% [55]. Current evidence regarding the association of PNR with OCT subtypes and treatment response remains limited. While conceptually attractive as a marker of thrombo-inflammatory balance, PNR requires validation in prospective and heterogeneous cohorts.

#### 5.1.7. Pan-Immune-Inflammation Value (PIV)

The PIV, calculated as (neutrophil × platelet × monocyte)/lymphocyte, has recently been advanced as a composite hematological index that captures neutrophil- and monocyte-driven innate immunity together with platelet-related thrombosis. Candan et al. conducted the first ophthalmology-focused evaluation of the role of PIV in DME and demonstrated an upward trend in PIV: 301 in healthy controls, 353 in diabetics without DR, and 452 in eyes with DME. A threshold of PIV > 427.7 effectively distinguished DME from healthy controls (sensitivity 81.7%, specificity 78.9%), while PIV > 451.3 differentiated DME from diabetics without DR (sensitivity 80.3%, specificity 77.5%) [56]. Both cut-off values are better than NLR and SII, suggesting that PIV may represent a promising novel biomarker for DME.

### 5.2. Platelet-Related Inflammatory Indices

#### 5.2.1. Mean Platelet Volume (MPV)

MPV, a marker of platelet activation and systemic thrombo-inflammatory status, shows a significant upward trend in DME; Tetikoğlu et al. reported that MPV in the DME group (8.87 ± 0.80 fL) was significantly higher than that in healthy controls (8.32 ± 0.9 fL) [57] and similar conclusions were drawn by Ilhan [51] and Mohamed [53]. A cohort study conducted in a Chinese population demonstrated the same trend with higher absolute values: controls 10.41 ± 0.63, DR without DME 11.07 ± 1.06 fL, and DR with DME 11.27 ± 0.85 fL [53]. Another study showed that MPV in DME patients with SRD (8.49 ± 1.77 fL) was higher than in those without SRD (8.05 ± 1.39 fL) [12]. These findings suggest that MPV may serve as a biomarker for the early diagnosis of DME.

The role of MPV in differentiating DME subtypes remains uncertain. Sun et al. found no significant differences in MPV across CME (11.39 ± 1.16 fL), DRT (11.54 ± 1.38 fL), and SRD (11.60 ± 1.20 fL) subtypes [55]. Conversely, Özata et al. reported elevated MPV in the SRD subtype [12]. Moreover, studies investigating MPV as a predictor of treatment response are limited and mostly retrospective. One study attempted to stratify patients with refractory DME undergoing dexamethasone implant treatment according to a baseline MPV threshold of 10.6, but no correlation was observed with improvements in BCVA or reductions in CRT [66].

The limitations of MPV include threshold dependency on specific cohorts and a lack of external validation. Furthermore, MPV is highly sensitive to pre-analytical variables (e.g., timing of measurement, type of anticoagulant, storage temperature), yet existing studies seldom describe standardization of these conditions. Finally, concomitant systemic diseases, smoking, or antiplatelet therapy may also compromise result accuracy.

#### 5.2.2. Platelet Distribution Width (PDW)

PDW, an index reflecting platelet heterogeneity and overall platelet mass, has also been investigated in DME. Li et al. observed a stepwise increase in PDW across the spectrum from healthy controls (11.93 ± 1.22) to diabetes (12.33 ± 1.28), DR (13.57 ± 2.25), and DME (13.89 ± 1.76). They suggested that PDW may reflect the inflammatory burden associated with retinal vascular injury [58]. However, other studies have not detected subtype-specific differences in PDW in DME. For instance, Zhu et al. reported that PDW in DME patients (16.28 ± 3.55%) was nearly identical to that in DR patients without DME (16.36 ± 6.86%; *p* = 0.933) [59]. Similarly, Sun et al. found PDW to be indistinguishable across CME, DRT, and SRD subtypes [55]. Tetikoğlu et al. noted only a non-significant upward trend in proliferative DR [57]. Taken together, the role of PDW in the diagnosis of DME and prediction of treatment outcomes remains to be clarified.

## 6. Evidence Grading and Clinical Classification

Based on a comprehensive review of observational studies identified from PubMed, Embase, and Web of Science, we evaluated the overall strength of evidence for each blood cell-derived inflammatory index in DME. Given the heterogeneity across studies with respect to populations, comparators, OCT-based DME subtypes, and outcome definitions, we followed the methodological guidance from the AHRQ Evidence-based Practice Center program and the GRADE Working Group [67,68], and graded the strength of findings by considering study design, sample size, clinical context (DME vs. non-DME, OCT-based subtypes, anti-VEGF treatment response), and the overall consistency of results. This approach allows readers to distinguish indices supported by relatively stronger observational evidence from those that remain in an exploratory stage. The detailed dataset, simplified risk-of-bias assessments, and methodological considerations are provided in Appendix A. Evidence was assigned to one of three categories—moderate, limited, and very limited—and each index was further classified as a core, secondary, or exploratory clinical marker, as summarized in Table 3.

## 7. Summary and Perspectives

The shifts in the epidemiology of diabetes have led to a growing prevalence of DME among younger individuals [69]. Compared with older adults, younger patients often exhibit a more inflammation-dominant phenotype with marked vascular permeability but with relatively mild diabetic retinopathy grades [70]. In addition, age influences the spectrum of comorbidities contributing to or mimicking DME. In younger individuals, highly prevalent myopia-related tractional pathologies and uveitis-associated edema can closely resemble DME on OCT imaging, while in the elderly population, macular diseases such as retinal vein occlusion and neovascular age-related macular degeneration are common confounders due to overlapping exudative features [71,72]. This emphasizes the importance of incorporating epidemiological shifts into clinical decision-making for DME.

Recent advances have highlighted inflammation as a central pathogenic driver. Blood-derived inflammatory biomarkers, such as NLR, PLR, SII, and MPV, have emerged as promising tools for predicting DME onset and therapeutic response. These indices are inexpensive, easily obtainable, and may aid in distinguishing between DME subtypes. More severe forms, including SRD and CME, often exhibit higher biomarker levels than DRT, suggesting potential utility in subtype stratification and diagnosis. In addition, they also correlate with treatment outcomes, with elevated levels consistently linked to poor response to anti-VEGF therapy. This offers a practical framework for identifying patients who may benefit from alternative or combination strategies.

However, it should be noted that most existing studies are single-center, retrospective, and small in scale, with heterogeneous populations and a lack of standardized diagnostic criteria for DME, which limits the generalizability of their findings. In addition, the possibility of publication bias cannot be excluded, as studies reporting significant associations between markers and DME may be more likely to be published than those with null results. Reported cutoff values for these markers also vary widely across studies, with no consensus on standardized thresholds, which may cause confusion in clinical application. Furthermore, patient characteristics such as age, obesity, and diabetes control can influence circulating inflammatory marker levels and act as potential confounders. These factors make the direct integration of these indices into routine DME management or treatment decision-making challenging, and their practical application requires further clinical validation.

Future research may address several key directions. Firstly, large-scale, multicenter, prospective studies with long-term follow-up across diverse DME populations are needed to validate the predictive value of blood inflammatory markers and establish standardized cutoff values. Secondly, these markers should be integrated with multimodal data, such as OCT imaging features and AI models, to develop comprehensive predictive models. For example, combining NLR and PLR with central macular thickness and cystoid area volume with machine learning frameworks could improve the accuracy of predicting anti-VEGF treatment response. This could shift DME management toward greater precision and personalization. Moreover, interventional studies will be important, such as evaluating whether controlling systemic inflammation through optimized diabetes management or anti-inflammatory agents can modify DME outcomes and clarify causal relationships between changes in biomarkers and disease progression.

## Figures and Tables

**Figure 1 biomedicines-13-02979-f001:**
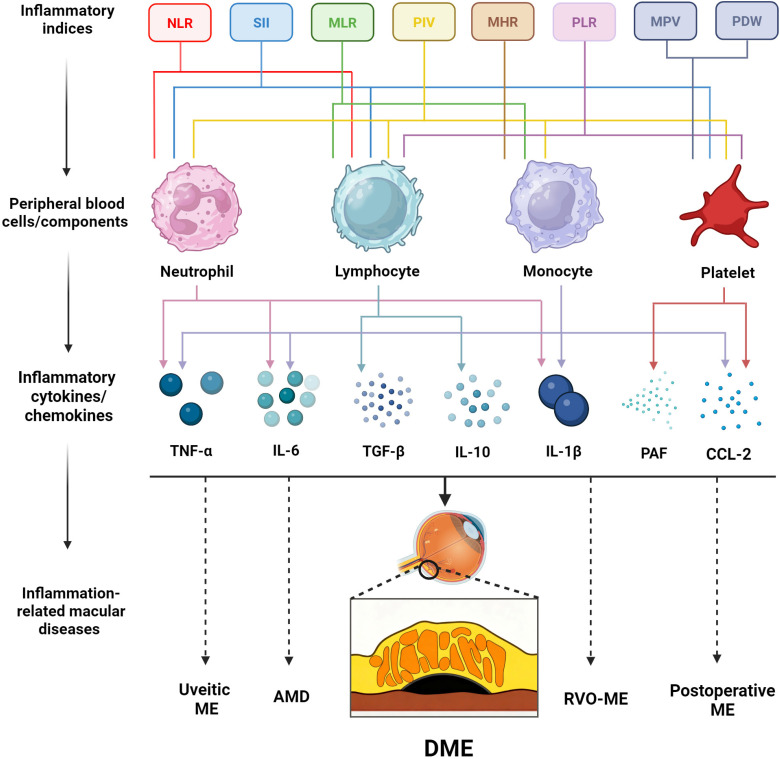
Conceptual framework illustrating blood-derived inflammatory indices, the peripheral blood cell populations and components they reflect, and associated inflammatory mediators, as well as their potential relevance to inflammation-related macular diseases, exemplified by DME. Created in BioRender. Der, B. (2025) https://BioRender.com/azrzn4x (accessed on 13 September 2025). NLR: neutrophil-to-lymphocyte ratio; SII: systemic immune-inflammation index; MLR: monocyte-to-lymphocyte ratio; PIV: pan-immune-inflammation value; MHR: monocyte-to-high-density lipoprotein cholesterol ratio; PLR: platelet-to-lymphocyte ratio; MPV: mean platelet volum; PDW: platelet distribution width; TNF-α: tumor necrosis factor-alpha; IL-6: interleukin-6; TGF-β: transforming growth factor-beta; IL-10: interleukin-10; IL-1β: interleukin-1 beta; PAF: platelet-activating factor; CCL-2: chemokine (C-C motif) ligand 2; ME: macular edema; DME: diabetic macular edema; AMD: age-related macular degeneration; RVO: retinal vein occlusion. Dotted lines indicate associations that remain hypothetical and require further investigation.

**Table 1 biomedicines-13-02979-t001:** Summary of studies and key findings on blood cell-derived inflammatory indices for DME detection and OCT-based subtypes classification.

Author and Year	Indices	Study Design	N	Diabetes Type	Control and Adjustment for Confounding Factors	Main Outcomes	Ref.
Özata Gündoğdu et al., 2022	NLR, PLR, SII, MPV	Retrospective	120	Not reported	Control: inflammation-related systemic diseases * No adjustment	The DME group shows a numerically higher but not statistically significant PLR. SRD subtype exhibits higher NLR, SII, and MPV values than those without SRD.	[12]
Liao et al., 2024	NLR, SII, MLR	Retrospective	50	Not reported	Control: inflammation-related systemic diseases * No adjustment	NLR ≥ 2.27, SII ≥ 447, and MLR ≥ 0.24 are diagnostic thresholds for CME.	[13]
Elbeyli et al., 2022	NLR, PLR, SII	Prospective	150	T2DM	Control: anti-inflammatory or Immunosuppressive drugs, inflammation-related systemic diseases * No adjustment	SII ≥ 399 is a diagnostic threshold for DME. The DME group demonstrates higher NLR, PLR, and SII values than DR-without-DME group.	[14]
Yalinbas Yeter et al., 2022	NLR, MHR	Retrospective, cross-sectional	143	T2DM	Control: anti-inflammatory drugs, fibrates or niacin, inflammation-related systemic diseases * No adjustment	NLR ≥ 2 and MHR ≥ 13.9 are diagnostic thresholds for DME.	[33]
Ilhan et al., 2020	NLR, MPV	Prospective	120	T2DM	Control: inflammation-related systemic diseases * No adjustment	NLR ≥ 2.26 is a diagnostic threshold for DME. Patients with DME demonstrate the highest MPV.	[51]
Mani et al., 2023	NLR, MPV	Case- control	114	T2DM	Control: anticoagulant or antiplatelet agents, smoke, inflammation-related systemic diseases * Adjustment: age	Elevated NLR and MPV are associated with DME.	[52]
Mohamed et al., 2024	PLR, MLR, MPV	Cohort	120	T2DM	Control: inflammation-related systemic diseases * No adjustment	The DME cohort exhibits elevated MLR, PLR, and MPV compared to diabetic patients without DME.	[53]
Tang et al., 2021	MHR	Cross-sectional	1378	T2DM	Control: anticoagulant or antiplatelet agents, inflammation-related systemic diseases * Adjustment: age, sex	MHR is similar between non-DME and DME cases.	[54]
Sun et al., 2025	PNR, MPV	Cross-sectional	366	T2DM	Control: antibiotics, immunosuppressants, anticoagulants, antiplatelet agents, steroids, or contraceptives, inflammation-related systemic diseases * Adjustment: DR severity	PNR ≤ 68.51 is a diagnostic threshold for DME. MPV fails to distinguish OCT-defined DME subtypes.	[55]
Candan et al., 2025	PIV	Case–control	155	Not reported	Control: anti-inflammatory drugs, inflammation-related systemic diseases * Adjustment: diabetes duration	PIV > 427.7 is a diagnostic threshold for DME. PIV > 451.3 separates DME from diabetes without retinopathy.	[56]
Tetikoğlu et al., 2016	MPV	Retrospective	199	Not reported	Control: antiplatelet agents, inflammation-related systemic diseases * No adjustment	The DME cohort exhibits elevated MPV compared with diabetics without DME.	[57]
Li et al., 2018	MPV, PDW	Retrospective	160	T2DM	Control: nephrotoxic agents or antiplatelet agents, inflammation-related systemic diseases * Adjustment: sex	The DME group demonstrates the highest MPV and PDW among healthy controls and diabetes without DME.	[58]
Zhu et al., 2022	PDW	Retrospective	114	T2DM	Control: inflammation-related systemic diseases * No adjustment	There is no difference in PDW between the DME and DR-without-DME group.	[59]

* Inflammation-related systemic diseases may include malignant tumors, liver and kidney dysfunction, autoimmune disorders, hematological diseases, coronary artery disease, and others. NLR: neutrophil-to-lymphocyte ratio; SII: systemic immune-inflammation index; MLR: monocyte-to-lymphocyte ratio; PIV: pan-immune-inflammation value; MHR: monocyte-to-high-density lipoprotein cholesterol ratio; PLR: platelet-to-lymphocyte ratio; PNR: platelet-to-neutrophil ratio; MPV: mean platelet volume; PDW: platelet distribution width; DME: diabetic macular edema; SRD: serous retinal detachment; CME: cystoid macular edema; T2DM: type 2 diabetes mellitus; DR: diabetic retinopathy; OCT: optical coherence tomography.

**Table 2 biomedicines-13-02979-t002:** Summary of studies and key findings on blood cell-derived inflammatory indices for predicting anti-VEGF response in DME.

Author and Year	Indices	Study Design	N	Diabetes Type	Control and Adjustment for Confounding Factors	Main Outcomes	Ref.
Katić et al., 2024	NLR, PLR, SII, MLR	Prospective	78	T2DM	Control: anti-inflammatory drugs or substitutional vitamin D therapy, inflammation-related systemic diseases * Adjustment: age, sex, C-reactive protein, disease duration	Lower baseline NLR, PLR, SII, and MLR predict a more favorable CRT outcome following anti-VEGF therapy.	[15]
Yalinbas Yeter et al., 2022	NLR, MHR	Retrospective, cross-sectional	143	T2DM	Control: anti-inflammatory drugs, fibrates or niacin, inflammation-related systemic diseases * No adjustment	Higher NLR predicts inferior CRT outcomes with anti-VEGF.	[33]
Hu et al., 2019	NLR, MLR	Retrospective	91	Not reported	Control: anti-inflammatory drugs, inflammation-related systemic diseases * No adjustment	Higher NLR contributes to inferior visual outcomes with anti-VEGF. MLR is negatively correlated with visual improvement following treatment.	[60]
Ergin et al., 2025	NLR, PLR, SII, MLR	Retrospective	104	T2DM	Control: anti-inflammatory drugs, inflammation-related systemic diseases * No adjustment	NLR ≤ 2.32, PLR ≤ 120.55, SII ≤ 543.53 and MLR ≤ 0.21 correlates with early anti-VEGF therapeutic response in DME.	[34]
Vural et al., 2021	NLR, MLR	Retrospective	64	T2DM	Control: inflammation-related systemic diseases * No adjustment	Elevated baseline NLR and MLR associate with a ≥1-line visual improvement following dexamethasone implantation.	[61]
Karimi et al., 2022	LNR, LPR	Prospective	80	Not reported	Control: inflammation-related systemic diseases * No adjustment	Higher baseline LNR and LPR are correlated with larger improvement in visual acuity after anti-VEGF therapy.	[62]
Chen et al., 2025	NLR, PLR	Retrospective	140	T2DM	Control: anti-inflammatory or immunosuppressive drugs, inflammation-related systemic diseases * Adjustment: age, Scr, eGFR, TG	NLR > 2.57 and PLR > 98.93 are predictive indicators of poor visual prognosis after anti-VEGF therapy.	[63]

* Inflammation-related systemic diseases may include malignant tumors, liver and kidney dysfunction, autoimmune disorders, hematological diseases, coronary artery disease, and others. NLR: neutrophil-to-lymphocyte ratio; SII: systemic immune-inflammation index; MLR: monocyte-to-lymphocyte ratio; MHR: monocyte-to-high-density lipoprotein cholesterol ratio; PLR: platelet-to-lymphocyte ratio; LNR: lymphocyte-to-neutrophil ratio; LPR: lymphocyte-to-platelet ratio; T2DM: type 2 diabetes mellitus; Scr: serum creatinine; eGFR: estimated glomerular filtration rate; TG: triglycerides; CRT: central retinal thickness; VEGF: vascular endothelial growth factor; DME: diabetic macular edema.

**Table 3 biomedicines-13-02979-t003:** Summary of evidence grading and clinical classification of blood cell-derived inflammatory indices in DME.

Indices	No. of Studies (n)	Prospective/Retrospective (n)	Studies Per Context * (DME vs. Non-DME/OCT-Based DME SubTypes/Anti-VEGF Treatment Response)	Total N per Study **, Median (Range)	ROC/AUC Studies (n)	AUC Range (Median) †	Evidence Grade ‡	Clinical Category §
NLR	12	3/9	9/4/3	120 (75–366)	7	0.551–0.800 (0.686)	Moderate (Grade B)	Core indices
PLR	10	3/7	7/3/3	120 (75–366)	4	0.586–0.719 (0.676)	Moderate (Grade B)	Core indices
SII	8	2/6	5/4/3	130 (75–366)	4	0.613–0.788 (0.696)	Moderate (Grade B)	Core indices
MPV	5	2/3	5/1/0	120 (120–160)	2	0.650–0.741 (0.696)	Limited (Grade C)	Secondary indices
MLR	7	2/5	4/1/3	120 (78–366)	3	0.565–0.704 (0.638)	Limited (Grade C)	Secondary indices
MHR	1	0/1	1/0/0	143 (143–143)	1	0.690–0.690 (0.690)	Very limited	Exploratory indices

* Number of independent studies evaluating each index in three predefined clinical contexts. ** Total sample size per study was approximated as the maximum total sample size (group 1 + group 2) reported for each index in each study. † AUC range and median among studies that reported ROC analyses for the given index. ‡ Evidence grades were assigned based on the domains of risk of bias, consistency, directness, and precision. § Clinical categories were pre-specified for this review and are not taken from any single guideline. NLR: neutrophil-to-lymphocyte ratio; PLR: platelet-to-lymphocyte ratio; SII: systemic immune-inflammation index; MPV: mean platelet volume; MLR: monocyte-to-lymphocyte ratio; MHR: monocyte-to-high-density lipoprotein cholesterol ratio; DME: diabetic macular edema; OCT: optical coherence tomography; VEGF: vascular endothelial growth factor; ROC: receiver operating characteristic; AUC: area under the curve.

## Data Availability

Not applicable.

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
