# Peer review of "Blood Cell-Derived Inflammatory Indices in Diabetic Macular Edema: Clinical Significance and Prognostic Relevance"

_biomedicines, 2025, doi:10.3390/biomedicines13122979_

Round 1
Reviewer 1 Report
Comments and Suggestions for Authors
Dear colleagues,
(1) Brief summary:
According to Diabetic macular edema (DME) is one of the most common cause of vision loss in patients with diabetes, which have become one of the epidemics of the XXI century, so that systematic reviews of biomedical research devoted to exploring the possibilities of the fastest, cheapest, and most reliable diagnostics for this disease have traditionally generated great interest among readers and, in this regard, undoubtedly deserve publication.
(2) Broad comments
The authors of this peer-reviewed, well-written, concise, easy-to-read, logical, and clearly understandable manuscript systematize clinical data on the most fastest, cheapest, and most reliable diagnostics based on blood components content tests available for diabetic macular edema (DME) in combination with optical coherence tomography as the cornerstone of modern non-invasive diagnostic techniques for retinopathies. Overall, the manuscript meets the requirements of the MDPI Author Guidelines, which is not surprising for authors with a case report article (PMID: 34533902) published in a journal classified with an impact factor of IF=78.5 and a quartile of Q1 according to the Web of Science Core Collection.
However, the manuscript under review is not without the following editorial deficiencies:
(3) Specific comments
(i) First of all, since diabetes and the resulting visual impairment due to diabetic macular edema (DME), which were traditionally considered age-related diseases (e.g., doi:10.1016/B978-0-443-15500-0.00012-8), are now rapidly becoming more prevalent in younger people, I believe the relevance and comprehensiveness of this manuscript could be significantly enhanced if the authors supplemented the revised manuscript, at least at the discussion and/or conclusion level, with information on the age-related characteristics of early DME diagnosis and its comorbidity with other diseases, especially those that can mimic DME symptoms, complicating its early diagnosis. I believe this is precisely what could be of greatest interest to a wider readership and, ultimately, increase the citation rate of the article after its publication.
(ii) The lack of graphical illustrations in a peer-reviewed manuscript may reduce the visibility of the relevant article to readers if it is published. This could be mitigated by supplementing the revised manuscript with a graphical illustration of their choice, such as, for example, the global incidence or first diagnosis of diabetes and/or DME from year to year, a scheme of DME comorbidity, and/or, conversely, its rare association with other pathologies. Otherwise, in my opinion, it is highly likely that readers will skim the relevant article, depriving themselves of the opportunity to learn the valuable information contained within.
(iii) In my opinion, Table 1 would look more understandable and logical if the authors could add another column "Ref" to the right, where they would move the reference numbers in square brackets from the first column as citations of literary sources.
(iv) References section: in reference #4 the authors are missing, namely:
the correct reference should be like this
GBD 2019 Blindness and Vision Impairment Collaborators; Vision Loss Expert Group of the Global Burden of Disease Study. Causes of blindness and vision impairment in 2020 and trends over 30 years, and prevalence of avoidable blindness in relation to VISION 2020: the Right to Sight: an analysis for the Global Burden of Disease Study. Lancet Glob Health. 2021;9(2):e144-e160. doi: 10.1016/S2214-109X(20)30489-7.
Author Response
Comments and Suggestions for Authors
Dear colleagues,
(1) Brief summary:
According to Diabetic macular edema (DME) is one of the most common cause of vision loss in patients with diabetes, which have become one of the epidemics of the XXI century, so that systematic reviews of biomedical research devoted to exploring the possibilities of the fastest, cheapest, and most reliable diagnostics for this disease have traditionally generated great interest among readers and, in this regard, undoubtedly deserve publication.
Response:
We sincerely thank the reviewer for the positive and encouraging summary of our work. We appreciate the recognition of the importance of this review in the field of diabetic macular edema and are grateful that our manuscript is considered valuable to the readership.
(2) Broad comments
The authors of this peer-reviewed, well-written, concise, easy-to-read, logical, and clearly understandable manuscript systematize clinical data on the most fastest, cheapest, and most reliable diagnostics based on blood components content tests available for diabetic macular edema (DME) in combination with optical coherence tomography as the cornerstone of modern non-invasive diagnostic techniques for retinopathies. Overall, the manuscript meets the requirements of the MDPI Author Guidelines, which is not surprising for authors with a case report article (PMID: 34533902) published in a journal classified with an impact factor of IF=78.5 and a quartile of Q1 according to the Web of Science Core Collection.
Response:
We are grateful for the reviewer’s comments. We appreciate the acknowledgement of the manuscript’s clarity, structure, and clinical relevance, as well as the recognition of our previous work. In response to the reviewer’s suggestions, we have revised and refined the manuscript to enhance its accuracy and overall quality.
However, the manuscript under review is not without the following editorial deficiencies:
(3) Specific comments
(i) First of all, since diabetes and the resulting visual impairment due to diabetic macular edema (DME), which were traditionally considered age-related diseases (e.g., doi:10.1016/B978-0-443-15500-0.00012-8), are now rapidly becoming more prevalent in younger people, I believe the relevance and comprehensiveness of this manuscript could be significantly enhanced if the authors supplemented the revised manuscript, at least at the discussion and/or conclusion level, with information on the age-related characteristics of early DME diagnosis and its comorbidity with other diseases, especially those that can mimic DME symptoms, complicating its early diagnosis. I believe this is precisely what could be of greatest interest to a wider readership and, ultimately, increase the citation rate of the article after its publication.
Response:
We sincerely thank the reviewer for this insightful comment. In the revised manuscript, we have added a new subsection in the Discussion to address the age-related characteristics of early DME diagnosis and its comorbidity profiles, particularly focusing on ocular diseases that may mimic DME and complicate its early detection (Page 13, Lines 372–381). We believe that this revision strengthens the clinical value and relevance of the manuscript by highlighting the shifting demographic trends of DME and the diagnostic challenges across different age groups.
(ii) The lack of graphical illustrations in a peer-reviewed manuscript may reduce the visibility of the relevant article to readers if it is published. This could be mitigated by supplementing the revised manuscript with a graphical illustration of their choice, such as, for example, the global incidence or first diagnosis of diabetes and/or DME from year to year, a scheme of DME comorbidity, and/or, conversely, its rare association with other pathologies. Otherwise, in my opinion, it is highly likely that readers will skim the relevant article, depriving themselves of the opportunity to learn the valuable information contained within.
Response:
We greatly appreciate the reviewer’s insightful suggestion that adding graphical illustrations can enhance the readability and overall appeal of a review article. In response, and considering the relevance to our research topic, we have included an original schematic figure in the revised manuscript. This figure presents the associations between various inflammation-related biomarkers and peripheral blood cells or inflammatory cytokines. In addition, it highlights the comorbidities of these biomarkers with other inflammation-associated retinal diseases, as well as potential future research directions (new Figure 1).
(iii) In my opinion, Table 1 would look more understandable and logical if the authors could add another column "Ref" to the right, where they would move the reference numbers in square brackets from the first column as citations of literary sources.
Response:
We thank the reviewer for this helpful suggestion. We have revised Table 1 accordingly by adding a “Ref” column and moving the reference numbers into that column.
(iv) References section: in reference #4 the authors are missing, namely:
the correct reference should be like this
GBD 2019 Blindness and Vision Impairment Collaborators; Vision Loss Expert Group of the Global Burden of Disease Study. Causes of blindness and vision impairment in 2020 and trends over 30 years, and prevalence of avoidable blindness in relation to VISION 2020: the Right to Sight: an analysis for the Global Burden of Disease Study. Lancet Glob Health. 2021;9(2):e144-e160. doi: 10.1016/S2214-109X(20)30489-7.
Response:
We thank the reviewer for the comments and now revise the reference for accuracy (ref #4)
Reviewer 2 Report
Comments and Suggestions for Authors
Lines 11–25: The abstract is too general and lacks quantitative evidence. Include representative diagnostic or prognostic values (AUC, sensitivity, specificity, or cut-off points) for key indices such as NLR or SII, and clarify whether conclusions are based on narrative or comparative evidence.
Lines 70–80: The therapeutic overview is disconnected from the biomarker focus.Briefly explain how inflammatory indices relate to treatment response, for example, that elevated SII may predict poor anti-VEGF response or better steroid responsiveness.
Lines 101–109: Table 1 lacks essential methodological information. Expand the table (or add supplementary material) to include study design, population characteristics, diabetes type, and whether confounders were controlled, to improve interpretability and comparability.
Lines 169–197: The discussion of SII mentions variable cut-offs but does not analyze the causes of heterogeneity. Add a short paragraph identifying possible sources of variability—sampling conditions, ethnic differences, comorbidities, or laboratory methods—and explain their impact on reproducibility.
Lines 198–234:Findings for MHR and MLR are inconsistent, and the section lacks synthesis. Provide a concise summary table or short paragraph grading the strength of evidence for each index (e.g., strong, moderate, inconsistent) to clarify interpretation.
Lines 293–312: The conclusion repeats earlier content and lacks clear research directions. : Replace repetitive statements with specific priorities such as standardization of cut-offs, multicenter validation, and integration of blood biomarkers with OCT features in predictive models.
The review is narrative but omits methodological transparency.Add a brief methods subsection describing the literature search strategy, databases, keywords, and time frame to ensure clarity and reproducibility.
Author Response
Comments and Suggestions for Authors
Lines 11–25: The abstract is too general and lacks quantitative evidence. Include representative diagnostic or prognostic values (AUC, sensitivity, specificity, or cut-off points) for key indices such as NLR or SII, and clarify whether conclusions are based on narrative or comparative evidence.
Response:
We thank the reviewer for this valuable comment. We have now added representative quantitative values for NLR in the abstract and clarified that these conclusions are based on comparative studies (Page 1 Line 18-22).
Lines 70–80: The therapeutic overview is disconnected from the biomarker focus. Briefly explain how inflammatory indices relate to treatment response, for example, that elevated SII may predict poor anti-VEGF response or better steroid responsiveness.
Response:
We agree with the reviewer’s suggestion. We have now added a paragraph explaining the relevance of inflammatory indices to treatment response. Specifically, accumulating evidence indicates that elevated baseline blood cell–derived inflammatory indices, such as the neutrophil-to-lymphocyte ratio (NLR), platelet-to-lymphocyte ratio (PLR), and systemic immune-inflammation index (SII), may predict poorer response to anti-VEGF therapy and a potentially better response to steroids [33,34] (Page 76 Line 81).
Lines 101–109: Table 1 lacks essential methodological information. Expand the table (or add supplementary material) to include study design, population characteristics, diabetes type, and whether confounders were controlled, to improve interpretability and comparability.
Response:
We appreciate the reviewer’s insightful comment. We have expanded the table and divided it into two separate tables, one summarizing studies evaluating the diagnostic value of inflammatory indices for distinguishing DME from non-DME and DME subtypes (new Table 1), and another summarizing studies investigating their prognostic value for treatment outcomes (new Table 2), which allow us to incorporate more methodological details. Specifically, we have added information on sample size, study design, diabetes type, and whether confounders were controlled.
Other characteristics may also include diabetes duration, baseline visual acuity and central macular thickness, and prior treatment history. We carefully reviewed the included studies and found that some of these variables were not consistently reported, they were therefore not incorporated into the tables.
Lines 169–197: The discussion of SII mentions variable cut-offs but does not analyze the causes of heterogeneity. Add a short paragraph identifying possible sources of variability—sampling conditions, ethnic differences, comorbidities, or laboratory methods—and explain their impact on reproducibility.
Response:
We thank the reviewer for this helpful suggestion. In accordance with the comment, we have added the following explanation to the revised manuscript: “This heterogeneity may be attributed to differences in study populations, ethnic backgrounds, comorbidities, DME morphologic subtypes, DR severity, anti-VEGF regimens, and response criteria across studies. Variability in sampling conditions and laboratory methods may also affect SII measurement. These factors can influence SII distribution and limit the reproducibility of a universal cut-off value across diverse clinical settings.” (Page 9, Line 237-242)
Lines 198–234: Findings for MHR and MLR are inconsistent, and the section lacks synthesis. Provide a concise summary table or short paragraph grading the strength of evidence for each index (e.g., strong, moderate, inconsistent) to clarify interpretation.
Response:
We thank the reviewer for this constructive suggestion. In response, we have incorporated evidence grading framework to synthesize the overall certainty of findings for each inflammatory index. Because all included studies were observational and showed substantial heterogeneity in study populations, comparators, OCT-based DME subtypes, and outcome definitions, a formal outcome-by-outcome GRADE assessment or quantitative meta-analysis was not feasible.
Following methodological guidance from the AHRQ Evidence-based Practice Center program and the GRADE Working Group, we pragmatically evaluated each index as a small “body of evidence” and assessed study-level limitations, consistency, directness, and precision. We then assigned an overall evidence grade (moderate, limited, or very limited) and a clinical category (core, secondary, or exploratory indices) to each biomarker (new Table 3).
This classification system and numerical thresholds were pre-specified for this review, not derived from a single existing guideline, and are intended to assist readers in distinguishing indices with relatively stronger evidence from those still in early exploration. The complete raw data (Supplementary information 2), simplified risk-of-bias evaluation (Supplementary information 3), and methodological details (Supplementary information 4) are provided in the supplementary materials.
Lines 293–312: The conclusion repeats earlier content and lacks clear research directions.: Replace repetitive statements with specific priorities such as standardization of cut-offs, multicenter validation, and integration of blood biomarkers with OCT features in predictive models.
Response:
We appreciate the reviewer’s suggestion and now replace the repetitive statements with the following content:
However, it should be noted that most existing studies are single-center, retrospective, and small in scale, with heterogeneous populations and a lack of standardized diagnostic criteria for DME, which limits the generalizability of their findings. In addition, the possibility of publication bias cannot be excluded, as studies reporting significant associations between markers and DME may be more likely to be published than those with null results. Reported cutoff values for these markers also vary widely across studies, with no consensus on standardized thresholds, which may cause confusion in clinical application. Furthermore, patient characteristics such as age, obesity, and diabetes control can influence circulating inflammatory marker levels and act as potential confounders. these factors make the direct integration of these indices into routine DME management or treatment decision-making challenging, and their practical application requires further clinical validation.
Future research may address several key directions. First, large-scale, multicenter, prospective studies with long-term follow-up across diverse DME populations are needed to validate the predictive value of blood inflammatory markers and establish standardized cutoff values. Second, these markers should be integrated with multi-modal data, such as OCT imaging features, to develop comprehensive predictive models. For example, combining NLR and PLR with central macular thickness and cystoid area volume, with machine learning frameworks could improve the accuracy of predicting anti-VEGF treatment response. This could shift DME management toward greater precision and personalization. Moreover, interventional studies will be important, such as evaluating whether controlling systemic inflammation through optimized diabetes management or anti-inflammatory agents can modify DME outcomes and clarify causal relationships between changes in biomarkers and disease progression. (Page 13 Line 391-414)
The review is narrative but omits methodological transparency. Add a brief methods subsection describing the literature search strategy, databases, keywords, and time frame to ensure clarity and reproducibility.
Response:
Thank you for this suggestion. We have now added a detailed description of the methodology, including the literature search strategy, databases, keywords, and time frame. The full details can be found in Supplementary information 2.

Reviewer 3 Report
Comments and Suggestions for Authors
Dear Editor
Thank you for providing me with the opportunity to review this manuscript.
This is a strong, high-quality review with immediate clinical relevance. Only minor methodological and interpretative refinements required.
1- While this is described as a “narrative review,” a clearer description of the literature search strategy, inclusion/exclusion criteria, and databases searched (PubMed, Embase, etc.) would improve transparency and reproducibility. Add a section "Literature Search" detailing databases (PubMed, Embase, Scopus), dates (inception–Oct 2025), terms (e.g., "DME AND (NLR OR PLR...)"), inclusion (human DME studies), yielding ~70 papers. State: "Narrative due to heterogeneity/small studies precluding meta-analysis."
2- There is a risk of bias in included studies. Most cited studies are retrospective and small-scale. The authors should explicitly discuss this limitation and potential publication bias.
3- The review does not adequately address how comorbid conditions (hypertension, infection, cardiovascular disease) or drugs (statins, antiplatelets) affect inflammatory indices. Include a summary table of potential confounders and their reported impact on these markers in DME populations. Tabulate confounders (age, HbA1c, nephropathy) from studies; subgroup by ethnicity (Asian vs. Caucasian differences in MPV?). Discuss antiplatelets' impact on PLR/SII.
4- Clarify the overlapping roles of indices (e.g., NLR vs. SII vs. PIV). A concise comparative table outlining mechanistic pathways (innate vs. adaptive immune contribution) would aid clarity.
5- The Table1 is excellent but dense. Consider separating diagnostic and prognostic findings or adding color/shading for readability.
6- Please review terminology, grammar and style. There are a few instances of minor spacing and punctuation inconsistencies (e.g., “be er” instead of “better,” likely OCR artifact). Use consistent phrasing for optical coherence tomography subtypes (“SRD,” “CME,” “DRT”). Ensure consistent abbreviation use; some terms (e.g., ML vs. MLR) appear inconsistently. Replace ambiguous expressions such as “we summarize on these biomarkers” with “we summarize the evidence regarding these biomarkers.”
7- There are some limitations that should be acknowledged in a limitation section in discussion:
- Predominance of single-center, retrospective studies with small cohorts.
- Lack of standardized cut-off values for biomarkers across populations.
- Potential confounding from systemic comorbidities (e.g., cardiovascular disease).
- Absence of prospective validation or interventional studies linking biomarker modulation to DME outcomes.
Author Response
Comments and Suggestions for Authors
Dear Editor
Thank you for providing me with the opportunity to review this manuscript.
This is a strong, high-quality review with immediate clinical relevance. Only minor methodological and interpretative refinements required.
Response:
We sincerely thank the reviewer for their positive evaluation and constructive feedback. We appreciate the recognition of our work’s quality and clinical relevance, and we have addressed the suggested methodological and interpretative refinements in the revised manuscript.
1- While this is described as a “narrative review,” a clearer description of the literature search strategy, inclusion/exclusion criteria, and databases searched (PubMed, Embase, etc.) would improve transparency and reproducibility. Add a section "Literature Search" detailing databases (PubMed, Embase, Scopus), dates (inception–Oct 2025), terms (e.g., "DME AND (NLR OR PLR...)"), inclusion (human DME studies), yielding ~70 papers. State: "Narrative due to heterogeneity/small studies precluding meta-analysis."
Response:
We sincerely appreciate the reviewer’s detailed and valuable comments. In response, we have now added a concise description of the methodology in the main text, and we have prepared a comprehensive methodological supplement that includes the literature search strategy, inclusion and exclusion criteria, databases searched, and search results. We also clearly stated that this review was conducted in a narrative format due to substantial heterogeneity among available studies and generally small sample sizes, which precluded a formal meta-analysis (Supplementary information 2).
2- There is a risk of bias in included studies. Most cited studies are retrospective and small-scale. The authors should explicitly discuss this limitation and potential publication bias.
Response:
We agree with the reviewer’s comment and have now addressed this limitation and the potential publication bias in the revised manuscript: However, it should be noted that most existing studies are single-center, retrospective, and small in scale, with heterogeneous populations and a lack of standardized diagnostic criteria for DME, which limits the generalizability of their findings. In addition, the possibility of publication bias cannot be excluded, as studies reporting significant associations between markers and DME may be more likely to be published than those with null results. Reported cutoff values for these markers also vary widely across studies, with no consensus on standardized thresholds, which may cause confusion in clinical application. Furthermore, patient characteristics such as age, obesity, and diabetes control can influence circulating inflammatory marker levels and act as potential confounders. these factors make the direct integration of these indices into routine DME management or treatment decision-making challenging, and their practical application requires further clinical validation. (Page 13, Line 391-402).
3- The review does not adequately address how comorbid conditions (hypertension, infection, cardiovascular disease) or drugs (statins, antiplatelets) affect inflammatory indices. Include a summary table of potential confounders and their reported impact on these markers in DME populations. Tabulate confounders (age, HbA1c, nephropathy) from studies; subgroup by ethnicity (Asian vs. Caucasian differences in MPV?). Discuss antiplatelets' impact on PLR/SII.
Response:
Thank you for these comments. We have now added information in the table on confounding control and confounder adjustment, as well as methodologic information such as study design and diabetes type. In addition, we have added a discussion about antiplatelet’s impact on PLR and SII in the revised manuscript (Page 8-9 Line 203-211, Page 9 Line 235-249).
4- Clarify the overlapping roles of indices (e.g., NLR vs. SII vs. PIV). A concise comparative table outlining mechanistic pathways (innate vs. adaptive immune contribution) would aid clarity.
Response:
We appreciate the reviewer’s suggestion. We have now added a new table regarding the immune mechanisms and pathways of peripheral blood-derived inflammatory indices in DME (Supplementary information 1).
5- The Table1 is excellent but dense. Consider separating diagnostic and prognostic findings or adding color/shading for readability.
Response:
We agree with the reviewer and now expanded the table and divided it into two separate tables, one summarizing studies evaluating the diagnostic value of inflammatory indices for distinguishing DME from non-DME and DME subtypes (new Table 1), and another summarizing studies investigating their prognostic value for treatment outcomes (new Table 2), which increase readability and allow us to incorporate more details.
6- Please review terminology, grammar and style. There are a few instances of minor spacing and punctuation inconsistencies (e.g., “be er” instead of “better,” likely OCR artifact). Use consistent phrasing for optical coherence tomography subtypes (“SRD,” “CME,” “DRT”). Ensure consistent abbreviation use; some terms (e.g., ML vs. MLR) appear inconsistently. Replace ambiguous expressions such as “we summarize on these biomarkers” with “we summarize the evidence regarding these biomarkers.”
Response:
We apologize for these errors. We have carefully reviewed the manuscript and corrected the inconsistent abbreviation, changing “ML” to “MLR,” and revised the phrase “summarize on these biomarkers” to “summarize the evidence regarding these biomarkers.” Regarding the reviewer’s note on the “be er” spelling error, we were unable to identify it; it may have arisen during file conversion. We sincerely appreciate the reviewer’s careful reading and valuable suggestions.
7- There are some limitations that should be acknowledged in a limitation section in discussion:
- Predominance of single-center, retrospective studies with small cohorts.
- Lack of standardized cut-off values for biomarkers across populations.
- Potential confounding from systemic comorbidities (e.g., cardiovascular disease).
- Absence of prospective validation or interventional studies linking biomarker modulation to DME outcomes.
Response:
We thank the reviewer for this insightful comment. In the revised manuscript, we have added a dedicated limitations section in the Discussion to clearly address the points raised, including the predominance of single-center retrospective studies, variability in biomarker cut-off values, potential systemic confounders, and the lack of prospective and interventional evidence (Page 13, Line 403-414).

Reviewer 4 Report
Comments and Suggestions for Authors
The review is timely and relevant, addressing a rapidly growing area linking systemic inflammatory indices with diabetic macular edema (DME).
Some comments follow:
The review would benefit from deeper analytical interpretation, not just listing findings. The authors should emphasize how these indices can practically influence clinical decision-making or future research directions.
Pathophysiology Section
Excellent overview of inflammation’s role in DME.
However, the text repeats similar mechanisms (cytokines, VEGF, oxidative stress) multiple times. Condense slightly and highlight which molecular pathways most strongly correlate with systemic indices (e.g., NF-κB–mediated neutrophil activation linked to NLR).
Adding a schematic figure showing this link would make this section more impactful.
Section 5 (Inflammatory Indices)
This is the core of the paper and is strong in detail. However:
Some subsections (e.g., NLR, PLR, SII) are overly repetitive in structure; condense by comparing and contrasting their predictive performance instead of repeating similar conclusions.
Include a comparative summary paragraph at the end of this section ranking indices by clinical promise or robustness (e.g., SII > NLR > PLR).
Discussion / Summary and Perspectives
Well-structured but slightly generic. Add:
A critical evaluation of whether these indices can realistically be integrated into routine DME management soon.
Discussion on cut-off variability and need for standardization.
Suggestions for future research, e.g., combining indices with OCT biomarkers or AI models.
A short paragraph on potential confounding factors such as age, obesity, or diabetic control that could alter inflammatory indices.
Minor stylistic points:
Replace “a ention” → “attention”.
No figures are included; a summary schematic or graphical abstract is strongly recommended.
Author Response
Comments and Suggestions for Authors
The review is timely and relevant, addressing a rapidly growing area linking systemic inflammatory indices with diabetic macular edema (DME).
Response:
We thank the reviewer for these comments. We are glad that the relevance and timeliness of our review are recognized.
Some comments follow:
The review would benefit from deeper analytical interpretation, not just listing findings. The authors should emphasize how these indices can practically influence clinical decision-making or future research directions.
Response:
We thank the reviewer for this valuable suggestion. We have added relevant content in the manuscript highlighting the current limitations of the studies and outlining specific directions for future research (Page 13, Line 391-414).
Pathophysiology Section
Excellent overview of inflammation’s role in DME.
However, the text repeats similar mechanisms (cytokines, VEGF, oxidative stress) multiple times. Condense slightly and highlight which molecular pathways most strongly correlate with systemic indices (e.g., NF-κB–mediated neutrophil activation linked to NLR).
Adding a schematic figure showing this link would make this section more impactful.
Response:
Thank you for these comments. We have reorganized and condensed the Pathophysiology section to avoid repetitive descriptions. Additionally, we have added a supplementary table to summarize and compare the predominant immune axes and putative mechanistic pathways associated with each inflammatory index (Supplementary Information 1).
Section 5 (Inflammatory Indices)
This is the core of the paper and is strong in detail. However:
Some subsections (e.g., NLR, PLR, SII) are overly repetitive in structure; condense by comparing and contrasting their predictive performance instead of repeating similar conclusions.
Response:
We have now reorganized these subsections to reduce redundancy and improve clarity. In addition, we have added a new Table 3 to provide a comparative evaluation of the importance and reliability of each systemic inflammatory marker.
Include a comparative summary paragraph at the end of this section ranking indices by clinical promise or robustness (e.g., SII > NLR > PLR).
Response:
We appreciate this insightful comment and now added a new section entitled “Evidence Grading and Clinical Classification” (Page 11-12 Line 342-356), along with a new Table 3, to provide a comparative evaluation of the importance and reliability of each systemic inflammatory marker.
Discussion / Summary and Perspectives
Well-structured but slightly generic. Add:
A critical evaluation of whether these indices can realistically be integrated into routine DME management soon.
Response:
We appreciate the reviewer’s valuable suggestion. In response, we have reorganized the Summary and Perspectives section to include a more critical evaluation of the feasibility of integrating these blood-derived inflammatory markers into routine DME management. Specifically, we now discuss the current challenges to clinical translation as well as future research directions, including the potential value of combining these biomarkers with OCT imaging features to improve predictive performance (Page 13, Line 403-414).
Discussion on cut-off variability and need for standardization.
Response:
We now clearly indicate the cut-off variability and need for standardization in the Summary and Perspectives section (Page 13 Line 391-402).
Suggestions for future research, e.g., combining indices with OCT biomarkers or AI models.
Response:
We now add future research direction in the Summary and Perspectives section, in which combining indices with OCT biomarkers or AI models are mentioned (Page 13 Line 407).
A short paragraph on potential confounding factors such as age, obesity, or diabetic control that could alter inflammatory indices.
Response:
Thank you for the comments. We have expanded the table and divided it into two separate tables, one summarizing studies evaluating the diagnostic value of inflammatory indices for distinguishing DME from non-DME and DME subtypes (new Table 1), and another summarizing studies investigating their prognostic value for treatment outcomes (new Table 2), which allow us to incorporate more methodological details. Specifically, we have added information on sample size, study design, diabetes type, and whether confounders were controlled.
Minor stylistic points:
Replace “a ention” → “attention”.
No figures are included; a summary schematic or graphical abstract is strongly recommended.
Response:
We appreciate these comments. We have carefully reviewed the entire manuscript. Regarding the reviewer’s note on the “a ention” spelling error, we were unable to identify it; it may have arisen during file conversion. In addition, we now add an original schematic figure in the revised manuscript. This figure presents the associations between various inflammation-related biomarkers and peripheral blood cells or inflammatory cytokines. In addition, it highlights the comorbidities of these biomarkers with other inflammation-associated retinal diseases, as well as potential future research directions.

Round 2
Reviewer 4 Report
Comments and Suggestions for Authors
Accept